# Overview of Tools and Measures Investigating Vaccine Hesitancy in a Ten Year Period: A Scoping Review

**DOI:** 10.3390/vaccines10081198

**Published:** 2022-07-27

**Authors:** Elizabeth O. Oduwole, Elizabeth D. Pienaar, Hassan Mahomed, Charles S. Wiysonge

**Affiliations:** 1Division of Health Systems and Public Health, Department of Global Health, Faculty of Medicine and Health Sciences, Stellenbosch University, Cape Town 7505, South Africa; hmahomed@sun.ac.za; 2Cochrane South Africa, South African Medical Research Council, Cape Town 7500, South Africa; epienaar65@gmail.com (E.D.P.); charles.wiysonge@mrc.ac.za (C.S.W.); 3Division of Epidemiology and Biostatistics, Department of Global Health, Faculty of Medicine and Health Sciences, Stellenbosch University, Cape Town 7505, South Africa; 4HIV and Other Infectious Diseases Research Unit, South African Medical Research Council, Durban 4091, South Africa

**Keywords:** vaccine hesitancy, immunization, vaccination, tools, measures, scoping review

## Abstract

The challenge of vaccine hesitancy, a growing global concern in the last decade, has been aggravated by the COVID-19 pandemic. The need for monitoring vaccine sentiments and early detection of vaccine hesitancy in a population recommended by the WHO calls for the availability of contextually relevant tools and measures. This scoping review covers a ten year-period from 2010–2019 which included the first nine years of the decade of vaccines and aims to give a broad overview of tools and measures, and present a summary of their nature, similarities, and differences. We conducted the review using the framework for scoping reviews by Arksey and O’Malley (2005) and reported it following the Preferred Reporting Items for Systematic reviews and Meta-Analyses extension for Scoping Reviews’ guidelines. Of the 26 studies included, only one was conducted in the WHO African Region. Measures for routine childhood vaccines were found to be the most preponderant in the reviewed literature. The need for validated, contextually relevant tools in the WHO Africa Region is essential, and made more so by the scourge of the ongoing pandemic in which vaccination is critical for curtailment.

## 1. Introduction

### 1.1. Background

The lifesaving impact of vaccination is again being brought into global focus, as it has become the primary preventive and containment measure available for the COVID-19 pandemic [1,2]. Prior to the outbreak of the pandemic, vaccination, which is an acclaimed successful public health intervention, was faced with various challenges. These ranged from pragmatic issues such as access and costs, to socio-behavioral issues such as vaccine hesitancy [3,4].

Vaccine hesitancy was previously defined as the delay in acceptance or refusal of vaccination services despite availability [5]. This definition was recently replaced in May 2022 by the one proposed by the World Health Organization (WHO) Behavioral and Social Drivers of Vaccination (BeSD) Working Group and endorsed by the WHO Strategic Advisory Group of Experts on immunization (SAGE). It is now defined as “a motivational state of being conflicted about, or opposed to, getting vaccinated; this includes intentions and willingness” [6]. Vaccine hesitancy exists on a continuum between those who accept all vaccines without doubt, to those who refuse all vaccines without doubt [5]. Vaccine hesitancy has impeded vaccination coverage and contributed significantly to the erosion of public health gains previously achieved by vaccination [7,8,9]. It has also been implicated in the resurgence of vaccine preventable diseases (VPDs) such as measles in various communities worldwide [7,9,10,11]. The context-specific nature of vaccine hesitancy and its variability across time, place and vaccines [5] makes vaccine hesitancy a complex problem to tackle and necessitates its investigation in diverse settings. This in turn, calls for the development of contextually relevant measures and tools. Vaccine refusal, the extreme expression of vaccine hesitancy is almost as old as the practice of vaccination itself [8,9], however, the term ‘vaccine hesitancy’ is relatively new, consolidated and defined by the SAGE in 2012 [5]. The growing challenge and questioning of vaccines and vaccination that had been smoldering for years reached crisis levels towards the end of the decade of vaccines (2011–2020) [12], prompting the WHO to declare it as one of the top ten threats to global health in 2019 [13]. The crisis soon became a raging inferno with the outbreak of the COVID-19 disease, caused by the SARS-CoV-2 virus. Declared a pandemic in March 2020 [14], several issues about vaccination against the infection inadvertently served as fuel for the “infodemic” [15,16,17,18] misinformation and disinformation that accompanied the pandemic. These issues include the relatively novel types of some of the then leading vaccine candidates [19], and the unprecedented rate at which they were tested and obtained emergency use authorization [19,20,21]. This further exacerbated the already existing problem of vaccine hesitancy in the pre-pandemic era. The pre-pandemic era was an era characterized by marked decline in vaccine confidence, demand and utilization by the general public [11,22,23,24,25,26] and even healthcare workers [27,28,29,30,31,32,33,34]. This was due to various factors, many of which were initially explained by the WHO SAGE ‘3C’ model of vaccine hesitancy of confidence, convenience and complacency [5,35]. This was later expanded by the inclusion of an additional two ‘Cs’ of rational calculation and collective responsibility [3,36]. However, neither these models nor the more complex matrix of vaccine hesitancy developed by SAGE [5,37] captures the breadth of vaccine hesitancy in all contexts, at all times and across all vaccines. Therefore, investigating different aspects of vaccine hesitancy in various contexts remains a priority issue. Moreover, vaccine hesitancy undermines vaccine demand and utilization, engendering conditions favorable to outbreaks and resurgence of VPDs; and eroding the gains of previous vaccination endeavors [9]. Like VPDs, therefore, vaccine hesitancy needs to be investigated, detected and eliminated or at least reduced to the barest minimum level possible in a community. Vaccination refusal which is the extreme expression of vaccine hesitancy has been around almost since the inception of modern vaccination practice by Edward Jenner in the late 1700’s [38,39], though its proponents were not as widespread nor as vociferous as they are now in recent times. Nevertheless, vaccination refusal and sub-optimal vaccination coverage and uptake has been investigated in various places by different methods, and the findings published accordingly. As VPDs gradually reduced, with many being eliminated in different parts of the world and one eradicated globally [40,41,42], the importance of vaccination seems to gradually diminish, and concerns about it increase. The rarity of many VPDs raised questions about the necessity of vaccination, breeding vaccination challenges anchored in psychosocial and behavioral factors. These types of challenges are different from the pragmatic challenges of vaccination such as issues of access and availability of vaccines. One such psychosocial challenges is vaccine hesitancy.

The context-specific nature of vaccine hesitancy and its variability across time, place and vaccines earlier mentioned, necessitates its investigation in different ways and in different settings. This has led to a plethora of studies investigating vaccine hesitancy, with some reporting on various tools and measures developed and/or adapted to investigate it. The method of development of these tools and measures varies considerably, making it difficult to compare findings across different settings, and by extension, impeding efficient tracking of variations in vaccine sentiments. This in turn impacts on the formulation of standard guidelines to ensure optimum vaccination up-take and compliance by global health authorities such as the WHO. The intention to mitigate this challenge was partly responsible for the development of the compendium of questions by the SAGE that was published by Larson et al. in 2015 [43]. This compendium of questions was recommended to be adapted and validated in different contexts with the aim of generating results that have a common basis for comparison. The literature accessed in the course of this review revealed that six studies (all of which are excluded because their parent tool is included in this review) were based on this compendium of questions. Further research will show if there are correlations among findings of studies based on the compendium of questions, and how useful or otherwise the tool is in generating comparable data that can be used to formulate a standard global vaccination monitoring and/or compliance guideline. 

To address the challenge of vaccine hesitancy, SAGE made three major categories of recommendations to the WHO, its partners and member states in its 2014 report [37]. The first category of the recommendations focused on the need to understand vaccine hesitancy, its determinants and the rapidly changing challenges that it entails. The second category addresses matters relating to structural and organizational capacities needed to decrease hesitancy and promote vaccine acceptance on global, national and local levels. The third recommendation prescribes the sharing of lessons learnt and best practices based on, and experience garnered in, various settings as well as the development, validation and implementation of new tools to address vaccine hesitancy [37,44]. This third recommendation informed the initial study of which this scoping review was a part, as detailed in the published protocol [9]. 

### 1.2. Review Rationale

Vaccine hesitancy was an emerging and growing threat to optimum vaccination coverage in the years prior to the outbreak of the COVID-19 pandemic, causing the WHO to, among other things, recommend the monitoring of vaccine sentiments and early detection of vaccine hesitancy in a population. This in turn calls for the availability of contextually relevant tools and measures, thus necessitating a review to scope the literature available at that period for such tools and their context of development and use. The review covered a period of ten years, from 2010 to 2019, a period which also included the first nine years of the decade of vaccines. The decade of vaccines (2011 to 2020) was declared by the World Health Assembly as part of the Global Vaccine Action Plan (GVAP) framework to achieve universal immunization coverage [45]. Also, it was during this period that the use of the term ‘vaccine hesitancy’ was consolidated, therefore, it can be safely assumed that tools designed specifically to measure it will begin to appear in literature published from this period. Prior to the writing of this review, few studies with similar concepts were published, an example of interest is the critical review conducted by Shapiro et al. in 2021 [46]. This critical review focused mainly on the methodology and psychometric properties of identified quantitative measures of childhood vaccine confidence between the years 2010 and 2019. Nevertheless, given the importance of vaccine hesitancy and its crucial role to the success or otherwise of vaccination endeavors, the need to scope available literature for tools and measures aimed at addressing it across different demography, vaccines and study designs remain relevant and important. This scoping review seeks to contribute to bridging this knowledge gap, and by including qualitative measures, highlight the potential and possibilities of conducting such investigations using qualitative methodology. Few reviews of this nature included such measures and studies. 

### 1.3. Review Objective 

The objective of this scoping review is to provide a broad overview of tools/measures addressing vaccine hesitancy published from 2010 to 2019, and highlight any point of interest about the study reporting them. This is to offer clinicians, researchers and any other interested parties a synopsis of what types of tools are available in the said period, in what populations were they developed and/or applied, which vaccines did they address, and where feasible, what results were obtained.

## 2. Materials and Methods

We applied the Askey and O’Malley framework for conducting scoping reviews [47] and incorporated suggested recommendations by other authors such as Levac, [48] Pham [49], and the Joanna Briggs Institute [50] and have reported the review following the PRISMA Extension for Scoping Reviews (PRISMA-ScR) guidelines [51]. The protocol [9] of this review was published a-priori. 

### 2.1. Eligibility Criteria

The primary eligibility criteria for studies to be included in this review is publication between the year 2010 and 2019. Studies were included if they contained or provided as Appendix A, tools (named, either validated or not) or measures (unnamed, but described in sufficient details to be considered to measure vaccine hesitancy or related concepts which can be used as a form of proxy to estimate the level of vaccine hesitancy in a population [12]). Relevant articles of either qualitative or quantitative study design, published in English, and in peer-reviewed journals were considered for inclusion in the review. Articles were excluded if they were not published within the specified time period of 2010–2019, if the included tools do not measure vaccine hesitancy or related concepts, or the tools were based on or are a modification or adaptation of a tool that is already included in the review. Grey literature and non-peer reviewed studies were also excluded.

### 2.2. Search Strategy

A modification of the three-step search strategy recommended by the JBI manual for review authors [50] was utilized in the development of the search strategy for this review, while the Askey and O’Malley framework for conducting scoping reviews [47] was the overall framework utilized in the conduct of the review. In the first step, an initial search of PubMed was conducted on the 6 June 2019 using broad search terms such as vaccine, vaccination, hesitant, hesitancy and refusal, to scan for articles of interest and to identify index terms and keywords used in such articles. The identified keywords and terms were used to formulate more comprehensive search strings in the second step of the search strategy. These were used to search PubMed, Web of Science and Scopus databases repeatedly to optimize the search strings. The final refined search strings were used to conduct the final search on the 11th of November 2019. The search included the PubMed, Web of Science, Scopus, and the EBSCOhost database in which the Cumulative Index to Nursing and Allied Health Literature (CINAHL), Africa Wide information, and Health source: Nursing/Academic Edition databases were searched. 

The searches were conducted by co-author Elizabeth D. Pienaar (EDP) (an information specialist), with the help of an experienced librarian in the Faculty of Medicine and Health Sciences library of Stellenbosch University. The scanning of a few selected articles by the PI (Elizabeth O. Oduwole (EOO)) for relevant studies did not yield any additional records. The final search strings including the Boolean operators used are detailed in the published protocol [9].

### 2.3. Study Selection

All records recovered from the final search of the databases indicated above were imported into EndNote reference manager where the initial deduplication was carried out by co-author EDP. The initial screening of titles and abstracts of the remaining records was conducted by lead author EOO, leading to the removal of more duplicates and the broad classification of records as either ‘irrelevant’, ‘include’ or ‘unsure’. Consultations with EDP assisted in the re-classification of the ‘unsure’ records into the two other categories. Studies classified as ‘include’ were subject to a second-round screening from which the final records to be included in the review were identified for full text screening. 

### 2.4. Data Charting

A data charting form designed specifically for the study was developed in Excel jointly by EOO and EDP with supervisory input from the two other authors. The refined form was pilot tested by EDP on four randomly selected articles, these served as a template for EOO who completed the data extraction process in conjunction with EDP. Information that was extracted included: year of publication, title, first author, country, WHO region, World Bank economic classification of the country in which the study was conducted, study type, name of tool, target population, vaccines investigated, domains/constructs investigated, number of items, item generation process, and if the tool was validated or not. This information is presented in Appendix A and presented descriptively and narratively in the following sections.

### 2.5. Data Synthesis and Analysis 

The data extracted from the full text of the final 26 studies eligible for inclusion in the review were collated and summarized narratively to give an overview of their nature, similarities and differences. 

## 3. Results

The final search of the six databases (PubMed, Web of Science, Scopus, CINAHL, Africa Wide information, and Health source: Nursing/Academic Edition) yielded a total of 12,300 records from which 4838 duplicates were removed. The exact number of records retrieved from each database and each step of the process leading to the final selection of the included studies are detailed in the PRISMA flow diagram in Figure 1.

Of the 21 excluded studies that were based on other tools (i.e., a variation, modification, translation or validation version of an original tool that was already selected for inclusion in the review); twelve studies were based on the Parent Attitudes about Childhood Vaccines (PACV) tool [52], six on WHO SAGE Working Group on Vaccine Hesitancy’s compendium of survey questions referred to in various publications as the Vaccine Hesitancy Scale (VHS) [43], two on the vaccination confidence scale (VCS) [53], and one on the Vaccination Attitude Examination (VAX) scale [54].

### 3.1. General Characteristics of Included Studies

Half (50%) of the included 26 studies were conducted in two countries in the WHO Region of the Americas: nine in the USA [29,52,53,54,55,56,57,58,59] and four in Canada [60,61,62,63]. The WHO European Region had four studies included (15%): two from the United Kingdom [64,65], one from Switzerland [43] and one from Germany though also tested in the USA [3]. The WHO Eastern Mediterranean Region had three (11%): one each from Saudi Arabia [66], Sudan [67] and Pakistan [68]. The WHO Western Pacific Region produced four (15%): two from Australia [69,70], one from China [71] and one from Hong Kong, Special Administrative Region of China [72]. Only one (4%) study was included from the WHO African Region, from Ghana [73]. One study (4%) was conducted in 67 countries selected from all six WHO regions [23]. Figure 2 illustrates this information graphically. The tools in the included studies were of two designs: 20 (77%) quantitative and 6 (23%) qualitative.

The study populations in which the tools or measures were developed or intended for use also differed, 15 of the studies targeted ‘parents’ of children of varying age groups and vaccination persuasions, six studies targeted general adult populations, pregnant women and post-partum women were the population of three studies, and one study named its population as ‘pediatric healthcare providers’. Expanded Program on Immunization (EPI) managers, health experts and related professionals were the study population of one study, another study named its population as healthcare professionals, health experts and frontline vaccination providers. One study reported a mixed population of parents, sick adults, health care workers and travelers in Germany as its tool development study population but applied it in a sample of general USA adults; and recommended the tool for use in the general adult population. 

A number of the studies (9/26) did not mention specific vaccines for their vaccine hesitancy investigating tools, while others (7/26) specified routine childhood vaccines. Other tools or measures in the included studies were aimed at specific vaccines such as human papilloma virus (HPV) vaccines (3/26), measles, mumps and rubella (MMR) vaccines (2/26), and different influenza virus (‘flu’) vaccines such as trivalent flu vaccines, childhood flu vaccines and seasonal flu vaccines (3/26) each. One tool investigated hesitancy for the pertussis and flu vaccines together [70], while another one mentioned that it investigated ‘adolescent vaccination’ [53]. 

Many of the included tools and measures (14/26) were reported to be validated though the methods and processes used differ and 11/26 did not indicate any form of validation. The analysis of the methodology of development and rigor of the reported validation processes are beyond the scope of this review. One tool, the vaccine hesitancy survey (VHS) [43] was recommended to be validated in different contexts.

### 3.2. Synopsis of Each Included Tool or Measure

In this review, the term ‘tool’ is used generally to refer to author-named measures investigating vaccine hesitancy or related concepts while the term ‘measure’ is used to refer to unnamed measures such as questionnaires, surveys and interview guides that are used to investigate or estimate vaccine hesitancy or related concepts. In a similar vein, an ‘item’ in this review refers to a question assessing or exploring a specific issue of interest on a measure or tool. The summary of the tools/measures are presented below based on their primary design.

#### 3.2.1. Quantitative Tools or Measures

Betsch and colleagues [3] developed and validated a tool in Germany and the USA to measure five psychological antecedents of vaccination, namely, confidence, constraints, complacency, calculation, collective responsibility (referred to as the 5C scale). The tool has a long form with 15 items and a short form with only five items. The tool is recommended by the authors to, among other uses, facilitate intervention design and global monitoring of any or all of the five named psychological antecedents of vaccination. A validated measure specifically assessing MMR vaccine hesitancy was developed and tested in the United Kingdom by Brown et al. [64]. The 27 itemed measure included 19 questions on attitudes, seven demographic questions, and one on previous behavior. The various responses to this single item were reported to be markedly and consistently different between MMR vaccine accepting and rejecting parents. 

A measure comprised of a total of 42 items of which 20 explored antenatal vaccination attitudes, intentions, social influences and risk considerations of pregnant women in Australia was used in a cross-sectional study conducted by Corben and Leask [69]. The study explored vaccination hesitancy in the antenatal period and found that majority of their respondents wanted their new baby to receive all recommended vaccinations, and that the likelihood to self-identify as unsure, somewhat, or very hesitant was thrice as high in first-time mothers than it is in other categories of mothers. Vaccine hesitancy was investigated among two panels of (a) Canadian health professionals, researchers, experts and policymakers and (b) front-line vaccine providers by means of pretested questionnaires consisting of a mixture of different types of questions including open ended questions, close ended questions, and binary response type of questions. The study [60]; “designed to assess the opinions of experts and health professionals concerning the definition, scope, and causes of vaccine hesitancy in Canada” as stated by the authors, reveals as part of its findings that most of the front-line vaccine providers included in the study support active listening of their clients vaccination concerns and providing accurate information in a nonjudgmental manner. It concludes that vaccine hesitancy is a concern for Canadian vaccination experts and health professionals. 

The Emory vaccine confidence index (EVCI) [59] is a validated tool developed based on vaccine confidence concepts as identified by the U.S. National Vaccine Advisory Committee. It contains, in its final form, a total of eight items, whose score could range from 0–24. These scores were calculated by summing scores from each collapsed variable and stratifying them into three-level categorical variables. Higher EVCI is associated with reported greater vaccine receipt in the test population, which by inference, indicates lower vaccine hesitancy for the routine childhood vaccine assessed. In 2014, Gilkey et al. [53] interrogated data from the National Immunization Survey—Teen conducted in 2010 using a three-factor, eight-item tool, the vaccination confidence scale (VCS). They found parents’ confidence in adolescent vaccination to be generally high, and recommended interventions that included heath literacy and cultural competencies to further bolster adolescent vaccination uptake in the US. An unnamed, though validated measure, with 15 items covering three sub-domains of ‘behavior’ (two items), ‘safety and efficacy’ (four items) and attitudes (nine items) was developed and tested in China by Hu and team in 2019 [71]. The authors recommend the survey for screening and identifying vaccine hesitant parents for targeted intervention aimed at increasing vaccine acceptance. 

The WHO SAGE working groups’ compendium of survey questions [43] incorporates the vaccine hesitancy survey (VHS), that is, ten items on a five-point Likert scale that has been validated in several contexts. The other component includes eleven core closed questions, five open-ended questions and four demographic questions. Developed by a rigorous process which included a thorough search of literature in a systematic review and extensive expert consultation and input, the compendium of questions was meant to be adapted and validated for use in different contexts to give a basis for comparing results of studies investigating vaccine hesitancy in varying settings. The vaccine confidence index (VCI) [23] is a brief, four-itemed tool that explored perceptions of vaccine importance, safety, effectiveness and compatibility with religious beliefs among 65,819 individuals in 67 countries across the globe. Notably the largest study on vaccination confidence conducted then, one of its’ key findings was that seven of the ten least vaccination confident countries are in the WHO European region. The study’s findings also reflected an inverse relationship between high socio-economic status and positive vaccine sentiments. 

The vaccination attitude scale (VAX scale) [54] is a validated measure of 12 items developed in a two-study process to assess general vaccination attitudes and explore their effectiveness in predicting future vaccination behavior. The authors propose its use to identify vaccination resisting individuals and their strongest objections to vaccination. The results of a cross-sectional survey conducted in Saudi Arabia [66] using a questionnaire containing an unspecified total number of items requiring mostly binary responses showed that slightly less than a third of the study population (n = 208) were generally vaccine hesitant, some were concerned about the safety of MMR vaccine, and some others had autism thoughts related to MMR vaccine. The study also reported a higher figure for age-appropriate fully immunized children than reported in other Arab and Muslim countries. One of the limitations of this study was the use of a self-reporting questionnaire, which predisposes the results to a high risk of recall bias.

The parent attitudes about childhood vaccines survey (PACV) [52] is the earliest tool developed specifically to investigate parental vaccine hesitancy about childhood vaccines in the United States. It is a validated tool consisting of a total of 18 items covering four content domains of: (a) immunization behavior (six items); (b) beliefs about vaccine safety and efficacy (eight items); (c) attitudes about vaccine mandates and exemptions (one item); and (d) trust (three items). It was developed through a process which included focus group discussions with vaccine hesitant parents (n = 4) and community pediatricians with vaccine hesitant parents as clients (n = 7). The authors conclude that the tool might be useful in identifying parents with immunization concerns for targeted interventions aimed at increasing childhood vaccine acceptance, but, nevertheless, recommended further psychometric evaluation. This tool is the most widely adapted/validated tool in different context, at least, 12 other studies (not included in this review according to its stated methodology) were based on it.

The HPV Attitudes and Beliefs Scale (HABS) [62] was developed and tested among a nationally representative sample of Canadian parents of boys aged 9–16 years, using data collected at two separate points in time. The validated tool consists of 46 items on 9-factor model solution and is said to be the first psychometrically-tested scale of HPV attitude and beliefs among parents of boys available for use in English and French by the authors who also recommended its’ further testing among parents of girls and young adults.

The vaccine acceptance instrument (VAI) [58] is a validated instrument containing 20 items in its full form and 10 items in its short form. Both forms of the tool contain items that address the five key facets of vaccine acceptance as described by the authors. The tool is recommended for use in understanding hesitant parents’ views and explaining variation in vaccine acceptance among the general population amongst other things. Developed and tested as a part of a two-wave longitudinal study, the vaccine conspiracy beliefs scale (VCBS) [63] consists of seven items with responses on a seven-point Likert scale ranging from strongly disagree to strongly agree. An example of an item in the tool is: “vaccine safety data is often fabricated”. The tool can be used to evaluate the impact of vaccine conspiracies beliefs on vaccine uptake and to further elucidate the correlates of vaccine hesitancy.

A purposively developed questionnaire of 23 questions exploring attitudes and practices of pediatric vaccine providers was used to investigate vaccine hesitancy among attendees of five American Academy of Pediatrics (AAP)-sponsored vaccine conferences in 2013 by Suryadevara and team [29]. The result of the study published in 2015 showed that almost all (661/666) of the participating attendees indicated that they routinely recommend standard pediatric vaccines. However, 30 of them admitted that they do not routinely recommend influenza and/or HPV vaccines. 

A cross sectional survey utilizing 55-itemed questionnaire was conducted among mothers who delivered within a 6-month time frame in Australia [70], to investigate antigen specific vaccine hesitancy during pregnancy. The results revealed that more of the 1014 participants who completed the pertussis section of the questionnaire accessed the pertussis booster dose during pregnancy, than did the 968 women who completed the influenza section and received the influenza vaccine. The caregiver vaccine acceptance scale (CVAS) [73] is a validated, 22-itemed tool developed and tested in Ghana, a lower-middle income country in the WHO Africa region. Nested in a regional survey whose primary purpose was to estimate regional vaccination coverage, the results showed high (≥80%) vaccine acceptance and compliance in the study population; while identifying varying percentages of caregivers expressing different concerns such as the number of vaccines administered in a single visit and the need for healthy children to be vaccinated. Fifteen percent of the caregivers admitted to having delayed or refused vaccination for reasons other than sickness or allergy. Developed and validated in Pakistan for use in low-income settings, the vaccine attitude scale (VAS) [68] consists of 14 items on a 5-point Likert scale. The tool contains two subscales; (a) vaccine perceptions and concerns (ten items) and (b) disease salience and community benefit (four items). The latter subscale was found to be the most concise, with the strongest association with childhood vaccination till the age of 4 months, and higher reliability than other scales identified by the authors. They recommended the four-item subscale for routine monitoring of parental attitudes and perceptions regarding childhood vaccination particularly in population with high burden of vaccine preventable diseases. 

An unnamed set of measures containing 20 items exploring different attitudes and behaviors of both general vaccine hesitancy and flu specific vaccine hesitancy was applied in a population of non-Hispanic black and white US-born, noninstitutionalized adults by Quinn et al. (2019) [57]. Their aim was to answer three research questions, one of which was: “how can we measure general vaccine hesitancy? How can we measure influenza specific vaccine hesitancy”? The study concludes amongst other findings that there is value in the utilization of general vaccine hesitancy and confidence measures, as well as vaccine specific measures. 

#### 3.2.2. Qualitative Tools/Measures

A qualitative study investigating reasons why women accept or reject the trivalent inactivated influenza vaccine during pregnancy was conducted by Meharry and team in 2010 [56] in a population of 60 women either in their 3rd trimester of pregnancy or had recently delivered healthy babies. Thematic analysis of the 16-itemed interview guide transcript revealed six themes, one of the key findings was that the “two-for-one” benefit is a pivotal piece of knowledge that influences future vaccination. 

The qualitative analysis of the written responses to the single question “Why didn’t you get the seasonal flu vaccine in the last flu season?” nested in a national survey conducted in a region of Ontario Canada by Meyer and Lum (2017) [61] provided evidence that supported the utility of the conceptual model of vaccine hesitancy (proffered by Dubé et al., (2013) [26]) for the design and analysis of research investigating seasonal flu vaccine refusal or delay. Most of the responses can be explained by the model, while a few could not because they were not related to vaccine hesitancy. 

In a study targeting vaccine hesitant parents (parents with a child overdue for a minimum of one vaccination for at least six months) conducted in the state of Utah in the US in 2010, the immunization hesitancy survey (IHS) [55] was developed and used by Luthy and co-workers. The survey consisted of five core open-ended questions and some demographic questions. Responses were coded and analyzed in a manner typical of qualitative study analyses. Two major themes emerged in the findings: (a) concerns regarding immunization safety and (b) lack of perceived need for immunization.

A qualitative enquiry into why some parents refused to vaccinate their children against influenza in the 2014/15 school pilot immunization program in England conducted by Paterson and team [65] reported among other findings that concerns about the vaccines’ safety and effectiveness, possible side effects and its constituents—such as porcine—to be largely responsible for their refusal.

With the goal to further understand measles vaccine hesitancy in Khartoum state, Sudan [67]; semi-structured interviews were conducted with five expanded programs on immunization (EPI) managers and nine frontline vaccination providers. The interview guide contained a mixture of binary (yes/no) response questions, open ended and demographic questions. Most of the study participants confirmed the existence of measles vaccine hesitancy, the main contextual influence identified was the presence of people referred to as “anti-vaccination” who belong to particular religious or ethnic groups in the population. 

The semi-structured interview guide used to explore the social and cultural construction processes involved in HPV vaccine hesitancy among a population (n = 40) of Chinese women in Hong Kong [72] consisted of a total of 39 questions: 28 interview guide questions and 11 demographic questions. The study found that factors such as cost of the vaccine, marriage plans, and experiences of sexual activities and history of experiencing gynecological conditions all influence the perception of HPV and HPV vaccines, which in turn, affects the receipt of HPV vaccines among the study population. Table 1 below presents a summary of the studies and tools/measures described above with the additional information of an example of each item from the tool/measure. A major advantage or findings of the study, tool or measure and a limitation of the study, or tool/measure development process as reported by the authors are also presented in the table. We noticed that there was no limitation reported for four of the studies included in the review by the authors.

#### 3.2.3. Major Similarities and Differences of the Tools and Measures

Previous studies have compared and contrasted different characteristics such as constructs and domains investigated [3,43], detailed psychometric properties examined and/or employed [46], and the number of scales/subscales [3,46] included in some of the tools included in this review. 

Other observed similarities include the method of tool development; most of the tools included in this review were developed basically by reviewing existing, relevant literature, by adapting items from existing tools and including some de novo items. The parent attitudes about childhood vaccines survey (PACV) [52] seems to be the favorite tool to validate, adapt, or use as a template to develop other tools since its publication in 2011. This may be because it is one of the earliest tools developed specifically to investigate parental vaccine hesitancy about childhood immunization. With the exception of the qualitative studies, most of the tools and measures consisted of one form of questionnaire or another, and almost all reported the inclusion of demographic questions. The domain or construct most commonly investigated where indicated is ‘attitude’. Others include concepts of trust, safety, effectiveness and accessibility all worded in different ways. Being developed and tested in high income countries was the most common similarity among the tools reviewed. 

A major observed difference in the reviewed tools is in the terminology used to describe the domains and/or constructs investigated. Different studies used terms such as attitude and behavior that have different meanings, to describe or refer to constructs explored by the same or similar items. Also, the categorization of items into scales and subscales differ for the tools reporting them, as does validation types and processes employed. The number of items also differed, ranging from a single item in one study [61] to as many as 55 in another [70]. A few of the tools reviewed included both a long and short form.

## 4. Discussion

The aim of this scoping review is to provide a broad overview of available vaccine hesitancy measuring tools in the first 9 years of the decade of vaccines, and to present a summary of their nature, similarities and differences.

Most of the tools/measures available in this period are quantitative in nature, less than a third of the included studies reported using qualitative measures. Admittedly, the reasons for conducting studies and the purpose of each tool differs; yet, it would have been gratifying to see more studies reporting the use of qualitative measures included in the review. Vaccine hesitancy is a psycho-behavioral issue, and by nature, a contextual phenomenon with variability across time, place and vaccines, attributes which seem to be better explored using qualitative measures. The time frame of our review may also have an influence on the retrieval of fewer studies reporting the use of qualitative measures. In the 10-year period covered by our scoping review, the term ‘vaccine hesitancy’ was relatively new, and the phenomenon an emerging threat to the success of vaccination endeavors. This period was before the outbreak of the COVID-19 disease, the advent of which brought major changes in the vaccination landscape, pushing vaccine hesitancy into the spotlight and elevating it from being just a health issue to a socio-political one. As incidence and effects of the pandemic gradually declines and the global community gradually returns to normality, it remains to be seen if there will be a change in how vaccine hesitancy is investigated and addressed. Possibly, there would be more studies reporting on the use of qualitative measures or at least incorporating more of its advantageous methods such as open-ended questions to investigate and address vaccine hesitancy.

The findings of our review confirm previous findings that most of the tools for investigating vaccine hesitancy were developed in high income countries [4,46] and precious few had been developed in low-middle income countries (LMICs). Of the 26 studies that met our inclusion criteria for this review, three were from LMIC setting [67,68,73]. Of these three, only the study of Wallace and team 2019 [73] was conducted in Sub-Saharan Africa. The study developed and tested the caregiver vaccine acceptance scale (CVAS) tool, and recommends it as a valid, contextually relevant tool to assess caregiver attitudes and beliefs towards vaccination in an LMIC setting and for monitoring vaccine confidence variations amongst other things. The other two studies conducted in LMIC settings are: (1) a qualitative study conducted in Sudan by Sabahelzain et al. (2019) [67] and (2) the quantitative study of Yousafzai et al. (2019) [68] conducted in Pakistan. Both countries are in the WHO Eastern Mediterranean Region (EMR). It is interesting to note that all the three studies conducted in LMIC settings were published in 2019, towards the tail end of the decade of vaccines. This evidence supports the theory that psychosocial and behavioral challenges, specifically vaccine hesitancy was still relatively new on the LMIC vaccination landscape in the pre pandemic era, but, nevertheless, was gaining ground [4]. It is expected that the scenario would have changed post COVID 19 pandemic; that more tools and measures aimed at vaccine hesitancy, specifically COVID 19 vaccine hesitancy would have been developed, validated and used in more LMICs settings. It is worthy of note that no studies from the WHO South-East Asia Region were included in this review. This could be due to the review’s inclusion criteria or possible paucity of relevant studies from the region.

Of the 23 tools and measures developed in high income countries (HICs) included in this review, three are of particular interest to us. The first is the WHO commissioned compendium of questions [43] earlier expanded on, the second is the parent attitudes about childhood vaccines survey (PACV) [52]. Developed, validated and tested in US by Opel and team, the PACV is the earliest, psychometrically validated, and most widely adapted tool specifically aimed at assessing parental vaccine hesitancy in various contexts since its publication in 2011. The PACV is reported to take less than 5 min to complete, read at a (US) grade 6 level and possess face and content validity. This tool is the most widely adapted and/or validated tool in different contexts of all available tools to measure vaccine hesitancy as earlier mentioned.

The third study of high interest is the state of vaccine confidence 2016: global insights through a 67-country survey [23]. Authored by Larson and colleagues, this is notably the largest study conducted in the pre-pandemic era investigating confidence in immunization in different contexts. Incorporating the vaccine confidence index (VCI) which is a brief four-itemed tool. The VCI was tested in among 65,819 individuals in 67 countries across all six WHO regions. This succinct tool targeted at monitoring variations in immunization attitudes at a global level, directly explores the complex and sensitive issue of vaccines and religious compatibility with one of its items. Few of the other tools included in this review explored this issue. Religion plays a crucial role in vaccination coverage and uptake [74,75,76,77], its influence is strong and bears direct or indirect association in the countries where wild polio virus is still circulating [74,78,79,80]. It is worth mentioning that a recent study estimating vaccine confidence levels among healthcare staff and students of a tertiary institution [12,81] conducted in South Africa by the authors of this review just before the roll out the COVID 19 vaccines for healthcare workers in South Africa, was based on the VCI tool.

Our study is not without limitations. As earlier mentioned, the relative newness of the term vaccine hesitancy limits the number of studies, and consequently tools specifically directed at measuring it. Heterogeneous terms used by different authors in the title of their studies may have led to some relevant studies with tools that did not assess vaccine hesitancy directly but measured related concepts such as vaccine confidence being missed. However, since most tools are developed drawing on extant literature, we are confident that the effect of this limitation is minimal. The restriction of language and non-inclusion of grey literature might also introduce some level of selection bias, but hopefully, the utility of this review still makes it advantageous in spite of this potential limitation.

## 5. Conclusions

The main objective of this scoping review was to provide a broad overview of vaccine hesitancy tools and measures available in the first 9 years of the decade of vaccines, which incidentally happens to be the pre COVID 19 pandemic era. The review also briefly highlighted their major similarities and differences, as well as their major advantages and disadvantages. Gaps in existing knowledge related to contextual development and validation of such tools, as well areas where more research is required were identified.

The synopsis of the 26 included tools/measures provides useful and relevant information for researchers of different levels and interests. Those intending to validate such tools in any context or setting will find the brief, cogent insights of the review a helpful resource.

We strongly recommend the development and validation of contextually relevant vaccine hesitancy measuring tools for all population sub-groups; and especially for COVID-19 vaccination in the WHO Africa region in particular, and LMIC settings in general.

## Figures and Tables

**Figure 1 vaccines-10-01198-f001:**
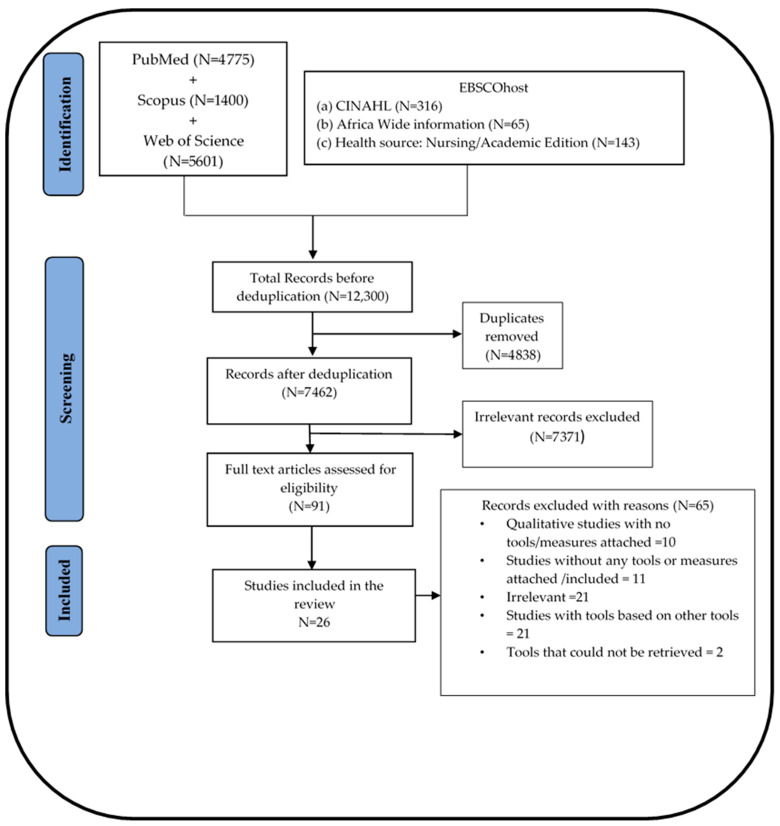
PRISMA flow diagram for the study selection process.

**Figure 2 vaccines-10-01198-f002:**
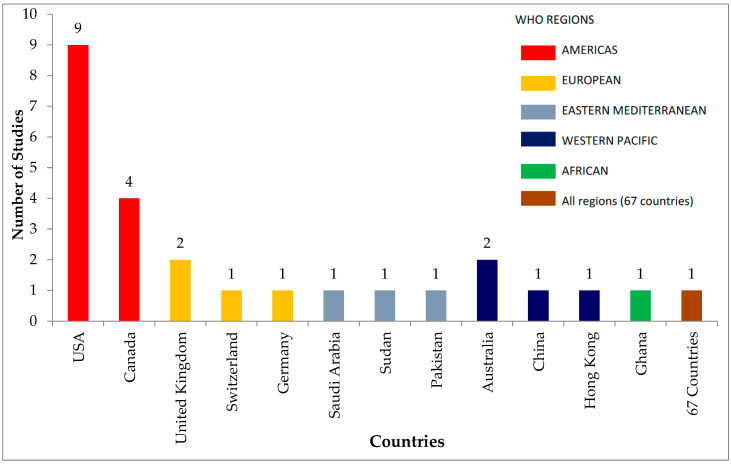
Number of included studies per WHO region and country.

**Table 1 vaccines-10-01198-t001:** Summary table with additional information.

First Author (Year)	Study Title	Example of Item	Major Advantage/Finding (as Reported by the Authors)	Major Disadvantage/Limitation (as Reported by the Authors)
Luthy, K.E. (2010)	Parental hesitation in immunizing children in Utah.	My child was delayed with immunizations because I have the following concerns about vaccine safety:	Hesitant parents have serious concerns regarding immunization safety.	The convenience sample was from one state, thus the findings cannot be generalized to other populations.
Brown, K.F. (2011)	Attitudinal and demographic predictors of measles, mumps and rubella (MMR) vaccine acceptance: Development and validation of an evidence-based measurement instrument.	MMR has serious side effects.	The instrument is able to elicit consistent responses at different time periods and on items which are conceptually/empirically linked, is able to discriminate between participants with differing MMR behaviors and is able to predict MMR behavior in multivariate analyses.	The study employed a retrospective design in which attitudes were measured after MMR doses were received, therefore the extent to which these attitudes incorporate retrospective justification and are able to predict future MMR behavior is yet to be ascertained.
Opel, D.J. (2011)	Development of a survey to identify vaccine-hesitant parents. The parent attitudes about childhood vaccines survey.	Have you ever decided not to have your child get a shot for reasons other than illness or allergy?	The final version of the PACV contains 18 items, takes 5 min or less to complete, and reads at a 6th grade level.	The total number of parents (N = 4) in the two focus groups was small and is therefore likely not representative of the larger vaccine hesitant parent population, a very heterogeneous group.
Meharry, P.M. (2012)	Reasons why women accept or reject the trivalent inactivated influenza vaccine (TIV) during pregnancy.	Who gave you advice about seasonal influenza and the influenza vaccine during your pregnancy?	The two-for-one benefit to mother and infant is pivotal knowledge and a predictor of future vaccination.	It was based on the experiences of 60 women during a specific time period, and although the sample had diverse cultural, educational and socio-economic backgrounds, the thematic analysis does not represent all possible responses.
Gilkey, M.B. (2014)	The Vaccination Confidence Scale: A brief measure of parents’ vaccination beliefs.	Vaccines are necessary to protect the health of teenagers.	Our nationally-representative sample allowed for the development of a robust tool tested with respect to demographic characteristics known to correlate with vaccination beliefs.	The primary limitation of this study was the modest number of items available for scale development.
Larson, H.J. (2015)	Measuring vaccine hesitancy: The development of a survey tool.	Childhood vaccines are important for my child’s health.	The Working Group developed a compendium of three different types of survey questions: core closed questions; Likert scale questions; and a set of open ended questions.	The questions identified do not address all the determinants in the Vaccine Hesitancy Matrix.
Suryadevara, M. (2015)	Pediatric provider vaccine hesitancy: An under-recognized obstacle to immunizing children.	Do you believe that standard immunizations are safe?	This is the first study to describe vaccine attitudes among pediatric providers attending AAP-sponsored immunization conferences.	The study population includes pediatric providers who attended AAP-sponsored conferences from a limited geographical area, and therefore the data may not be generalizable to all pediatric providers.
Eve Dubé (2016)	Understanding Vaccine Hesitancy in Canada: Results of a Consultation Study by the Canadian Immunization Research Network.	How prepared are you to effectively provide information about risks and benefits of vaccination?	Our findings indicate that the majority of participants—both vaccine experts and front-line vaccine providers—have the perception that vaccine rates have been declining and consider vaccine hesitancy an important issue to address in Canada.	By design, the results reported here represent the opinions of only some non-randomly selected key opinion leaders.
Larson, H.J. (2016)	The State of Vaccine Confidence 2016: Global Insights Through a 67-Country Survey.	Vaccines are compatible with my religious beliefs.	We find that vaccine safety sentiment is more negative in the European and the Western Pacific regions, where nine of the ten least confident countries are located.	A limitation of this survey is its generality of the survey which does not reveal whether the attitudes are related to specific vaccine(s) which an individual may have concerns about.
Perez, S. (2016)	Development and Validation of the Human Papillomavirus Attitudes and Beliefs Scale in a National Canadian Sample.	I feel that… the HPV vaccine will protect my son’s sexual health.	The HABS is available in both English and French for assessing HPV attitudes and beliefs.	The HABS does not capture all attitudinal items.
Shapiro, G.K. (2016)	Validation of the vaccine conspiracy beliefs scale.	Immunizing children is harmful and this fact is covered up.	Income, parental age, healthcare provider recommendation, and vaccine conspiracy beliefs emerged as significant predictors of parents’ willingness to vaccinate their child.	Using preselected items to develop this scale may have increased the likelihood of producing a one-dimensional scale.
Martin, L.R. (2017)	Understanding the Dimensions of Anti-Vaccination Attitudes: the Vaccination Attitudes Examination (VAX) Scale.	Vaccines can cause unforeseen problems in children.	The VAX scale is a short and simple tool that has demonstrated significant associations with vaccination behaviors and intentions.	No study limitation was reported by the authors of this study.
Meyer, S.B. (2017)	Explanations for Not Receiving the Seasonal Influenza Vaccine: An Ontario Canada Based Survey.	Why didn’t you get the seasonal flu vaccine in the last flu season?	The most cited explanation given for not receiving the seasonal influenza vaccine is related to the perceived importance of vaccination (or lack thereof).	Due to the nature of our data collection, we were unable to continue to sample until reaching saturation of themes.
Betsch, C. (2018)	Beyond confidence: Development of a measure assessing the 5C psychological antecedents of vaccination.	For me, it is inconvenient to receive vaccinations.	The 5C scale now offers a psychologically sound and validated measure to be used for regular global monitoring of the psychological antecedents of vaccination behavior.	A limitation of this work is that the three studies, similar to the construction studies of all other existing measures, only assess concurrent validity and not predictive validity.
Paul Corben (2018)	Vaccination hesitancy in the antenatal period: a cross-sectional survey.	All things considered, how much do you trust your child’s doctor?	There was no difference detected in vaccination timeliness of babies of first-time mothers and experienced mothers nor between those who considered themselves ‘not at all hesitant’ and others.	We were unable to calculate summary hesitancy and decisional conflict measures.
Paula M. Frew (2018)	Development of a US trust measure to assess and monitor parental confidence in the vaccine system.	Vaccines recommended for young children are safe.	[The authors] developed a parsimonious, relevant eight-item index that was able to assess vaccine confidence with a highly acceptable internal validity score.	Several sources of bias limit the ability of self-reported vaccination decisions to represent actual vaccination behavior, including recall, response, and social desirability bias.
Paterson, P. (2018)	Reasons for non-vaccination: Parental vaccine hesitancy and the childhood influenza vaccination school pilot programme in England	‘‘We would be grateful if you could tell us why you decided not to vaccinate your child as part of the school immunization program”	The majority of parents interviewed illustrated a lack of perceived need for the influenza vaccine for children.	Study limitations include the possibility of sample bias, since those that took part in our study might have different views to that of the general population.
Sarathchandra, D. (2018)	A survey instrument for measuring vaccine acceptance.	My right to consent to medical treatment means that vaccinations should always be voluntary.	Our results indicate that vaccine acceptance is substantially eroded by conspiratorial thinking and is modestly reduced by political conservatism.	No study limitation was reported by the authors of this study.
Hu Yu (2019)	Reliability and validity of a survey to identify vaccine hesitancy among parents in Changxing county, Zhejiang province.	It is better for my child to develop immunity by getting sick than vaccination.	We found the concern of the vaccine efficacy was associated with under immunization, which was similar to the previous studies in other settings.	Our results might reflect current perceptions of immunizations other than perceptions at the time they were making immunization decisions,
Mohamed, M.M. (2019)	The prevalence of vaccine hesitancy and skipping MMR vaccine due to autism thoughts in Saudi Arabia.	Do you think that your child received too many vaccines?	Vaccination hesitant parents showed a significantly high probability that they think that healthy children don’t need to be vaccinated with MMR, and that the risk of MMR vaccine outweighs the benefit.	One of the limitations of this study was using self-reported questionnaires for collecting data which were prone to recall bias.
Quinn, S.C. (2019)	Measuring vaccine hesitancy, confidence, trust and flu vaccine uptake: Results of a national survey of White and African American adults.	Thinking specifically about the flu vaccine, do you think the flu vaccine is necessary?	In this article, we can distinguish between general vaccine hesitancy and vaccine hesitancy specific to the flu vaccine.	No study limitation was reported by the authors of this study.
Sabahelzain, M.M. (2019)	Towards a further understanding of measles vaccine hesitancy in Khartoum state, Sudan: A qualitative study.	Do you think measles vaccine hesitancy exists in Sudan? Why?	The majority of the participants agreed that the main contextual determinant is the presence of people (parents/guardians) who can be qualified as “anti-vaccination”; they mostly belong to religious groups, and they often refuse all vaccines.	This study’s findings should be interpreted within the context of the study ’s participants and areas.
Siu, J.Y.M. (2019)	Social and cultural construction processes involved in HPV vaccine hesitancy among Chinese women: a qualitative study.	How do you perceive the dangers of HPV?	Only a few participants knew that HPV could lead to genital warts and that HPV vaccination can also help prevent genital warts.	Our findings mostly reflect the perceptions and decision-making process of women who belong to a relatively high socioeconomic status.
Van Buynder, P.G. (2019)	Antigen specific vaccine hesitancy in pregnancy.	During which trimester of this pregnancy did you receive the flu vaccine?	One out of every two pregnant women surveyed accessed a pertussis vaccine booster but not an influenza vaccine.	No study limitation was reported by the authors of this study.
Wallace, A.S. (2019)	Development of a valid and reliable scale to assess parents’ beliefs and attitudes about childhood vaccines and their association with vaccination uptake and delay in Ghana.	People in this community have expressed concerns that a child might have a serious side effect from a vaccination.	Our study is the first to document development of a valid and reliable scale to assess caregiver attitudes and beliefs towards vaccination in a low- or middle- income country setting and show a high level of association of the scale score with child’s vaccination status.	The survey was cross-sectional, so information for the scale and for vaccination status was collected at the same time; thus, our criterion validity was limited to concurrent rather than predictive validity.
Yousafzai, M.T. (2019)	Development and Validation of Parental Vaccine Attitudes Scale for Use in Low-income Setting.	I should be allowed to selectively choose the vaccines which I believe my child needs.	The four-item scale addressing parental attitude toward vaccine-preventable disease salience and community benefit is sufficiently reliable, and it can predict vaccine acceptance among parents in low-income settings.	One of the limitations of this study is validation of the tool only in Pakistan, rather than in several developing countries.

## Data Availability

All data relevant to the study are included in the article.

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
