# Peer review of "Overview of Tools and Measures Investigating Vaccine Hesitancy in a Ten Year Period: A Scoping Review"

_vaccines, 2022, doi:10.3390/vaccines10081198_

Round 1

Reviewer 1 Report

The paper reports a scoping review of tools and measures investigating vaccine hesitancy in a ten year period of 2010-2019. The topic is relevant and timely. My concerns are mostly related to the selection of papers included in the review and on the way those papers have been discussed in the Results.

I list my comments below:

-          Line 22: the sentence “the final searches were…” is too detailed for the Abstract; my suggestion would be to delete the sentence from the Abstract;

-          Line 23: “was conducted IN IN the WHO…”; “in “ is repeated twice;

-          Line 40: “as earlier mentioned”; where was it mentioned?

-          The expression “decade of vaccines” has been used throughout the paper but it is not clear to me what the Authors mean; please, define;

-          Line 101: the review by Shapiro is mentioned but the reader would appreciate to read some more lines about differences and similarities between Shapiro’s and the Authors’ work; In what your paper is new?

-          Line 105-107: please, expand the objectives of the review; why is your work relevant? In what it is new?

-          About the selection of studies: both quantitative and qualitative tools/measures were selected; to me, assess means quantitatively measure a process/phenomenon, and in this sense only quantitative studies (involving questionnaires or validated scales) should be included into the review; readers could proficiently validate a questionnaire/scale into a new country/context, while the process of validation a qualitative tool (such as interview guides) could be more discretionary. Also, qualitative studies are often reliable but pose some problems as for the replication. My suggestion would be to exclude the qualitative studies from the review. If the Authors disagree with this, please provide an explanation for your choice.

-          Line 141 onward: the name of the authors are not necessary; initials are just enough.

-          Line 162: in the Data charting (or in the Results section), the reader would benefit from knowing which are the mostly used tools, if they were translated/validate in which language and so on. A tool used by more than 10 other studies is somehow more sound than a tool used only once. I think that exploring this via Google Scholar “cited by” option would be feasible.

-          In the Results section, the reader would appreciate to know the major pros and cons for each tool; for instance, one tool has been translated and used many times, whereas another one has been used only once; one tool is very long and time-consuming, whereas the other is shorter, and so on.

-          In the 3.2.1 section: while describing the studies, the main findings are also reported in terms of percentage of people supporting (or not supporting) vaccine; I am not sure this kind of data are relevant for the aims of the paper. If the paper aims to “provide a broad overview of tools…” focusing on the methods used, then the results of single studies are not relevant. Moreover, as non-representative convenience samples are involved very often in this kind of studies, then the percentage of respondents supporting (or being hesitant towards) vaccines is not a reliable measure of vaccine hesitancy in that country; that figure only describes the recruited sample. I would suggest to delete those information.

-          The reader would appreciate to have an example of an item for each tool; in some case, it is reported such as line 325-326, in other case it is not.

-          Other relevant information that the reader would appreciate are the n. of items for each measure, the response scale (for quantitative measures only), how often has been used (often, rarely etc etc), if it has been validated or not; those data should be put in the Data Charting (Excell file) and/or in the Results section when describing each tool.

-          Line 406-409: I am not sure whether those lines are relevant for the aims of the paper;

-          In the Discussion section, lines 488-512: as those are the results of the scoping review, I would personally prefer to see those precious comments in the Results section; in other words, the reader would appreciate a reasoned review, where the Authors provide some insightful comments on the tools they have selected; I think this is a valuable result of the study.

-          I have appreciated very much the call for a development of validated measures of vaccine hesitancy in WHO Africa region, as suggested in lines 534-537.

Reviewer 2 Report

This is a well-written and well-presented study.  My only concerns are the depth of the discussion and conclusions.  The first 2-3 full paragraphs in the discussion belong in Introduction's background section.  I think that the discussion should be more detailed and draw on the findings.  The conclusion is anti-climactic but it could be so much more and I think the authors are capable of bringing more insight into this section.

Reviewer 3 Report

I felt the paper was certainly worthy of publication my only two comments would have been: 1) to remove the name of one of the participants from the methods, just de-identify the person; 2) instead of describing the % in the Result section, use a graph to show these results.

Reviewer 4 Report

The literature review proposed is interesting and current. It also describes how vaccine hesitation has changed in the era of the pandemic and infodemic.

Please make a note of the following tips.

Introduction: No reference was made to the general concern about vaccination hesitation of healthcare professionals (i.e. doi: 10.3390/vaccines9070713; doi: 10.3390/vaccines10060948) in particular in the future perspective (viruses co-circulation and vaccination policies DOI: 10.1016/j.jvacx.2022.100172).

Results: please reflect on the importance of a possible summary table, it would greatly help the reader who does not go to read the supplementary material (see who reads from smart-phones).

Discussion: focuses heavily on the past. It should address more the problem of infodemic and its influence (doi: 10.2196/38423; doi: 10.2196/38034). Some ideas on how to deal with the problem should also be included.
